# Performance and muscle lipogenesis of calves born to Nellore cows with different residual feed intake classification

**Ana Carolina Almeida Rollo de Paz**[1], **Márcio Machado Ladeira**[2], **Priscilla Dutra Teixeira**[2], **Roberta Carrilho Canesin**[1], **Camila Delveaux Araujo Batalha**[1], **Maria Eugênia Zerlotti Mercadante**[1], **Sarah Figueiredo Martins Bonilha**[1]*

1 Instituto de Zootecnia, Centro Avançado de Pesquisa e Desenvolvimento de Bovinos de Corte, Rodovia Carlos Tonani, Sertãozinho, São Paulo, Brazil, 2 Universidade Federal de Lavras, Departamento de Zootecnia, Lavras, Minas Gerais, Brazil

* sarah.bonilha@sp.gov.br

**Data Availability Statement:** All relevant data are within the manuscript and its Supporting information files.

## Abstract

This study aimed to evaluate relationships among maternal residual feed intake (RFI) with growth performance and expression of genes involved in lipid metabolism in offspring of Nellore cattle. Fifty-three cows classified as negative or positive RFI by genomic prediction were exposed to fixed-time artificial insemination (FTAI) protocols at 2 and 3 years of age using semen from the same bull. In the first year, cows gestated under grazing conditions and nursed their calves in the feedlot. In the second year, the opposite occurred. Cows were weighed every 28 days during pregnancy and calves were weighed at birth and every 28 days until weaning. Ultrasound images were collected from the carcass of cows and calves. Muscle gene expression was evaluated in calves at birth and weaning. Data were analyzed by year considering the fixed effects of RFI class and FTAI protocol for variables measured in cows, and RFI class, FTAI protocol and sex for variables measured in calves. There was no effect of maternal RFI on calves performance in the first year. Lower expression of *FABP4* gene and trend towards lower expression of *SREBF1* and *LPL* genes were detected in samples collected after birth from calves born to negative RFI cows, indicating that adipogenesis was reduced during the fetal and neonatal period. In the second year, negative RFI cows had greater subcutaneous fat thickness than positive RFI cows, and their calves tended to be heavier at birth and to have less rump fat thickness at weaning. No significant differences in expression of genes studied were detected between cow RFI classes. Nellore cows classified as negative RFI consume less feed and produce calves whose growth potential is similar to that of calves produced by positive RFI cows.

## Introduction

Calf production is an important step for beef cattle system, being the first stage towards efficient livestock production. Maternal nutrition during pregnancy and lactation is crucial for the pre- and postnatal development of the offspring [1]. The low nutrient availability during

**Funding:** This work was supported by São Paulo Research Foundation (FAPESP): grant #2017/06709-2 provided to SFMB; and #2018/20080-2 provided to CDAB. This work was also supported by Coordination for the Improvement of Higher Education Personnel (CAPES, Finance Code 001), providing grant to ACARP. The funders had no role in study design, data collection and analysis, decision to publish, or preparation of the manuscript. There was no additional external funding received for this study.

**Competing interests:** The authors have declared that no competing interests exist.

the second trimester of gestation can have long-term effects on growth, body composition and metabolism of the offspring, affecting the development of fetal body tissues [2,3]. Furthermore, the maternal diet can affect the long-term expression of genes through epigenetic mechanisms since the nutritional stimulus is recorded by the genome and revealed by gene expression [4]. In other words, nutrient restriction during the fetal period can result in impaired growth rates and productivity throughout the offspring's productive life [5]. Understanding this process and all factors involved is therefore important to improve the production indices of beef cattle herds.

The identification of feed efficient animals is also a key factor for the productive efficiency of beef cattle farming. It can be used as a strategy to control the costs and enhance sustainability of production systems. Residual feed intake (RFI) is a feed efficiency measure that explores variations in the feed intake of animals and is defined as the difference between observed and predicted dry matter intake as a function of the rate of gain and metabolic body weight (BW) of the animal [6]. Studies comparing animals classified as more or less feed efficient based on RFI have shown differences in the efficiency of energy utilization for maintenance [7]. Animals with lower net energy requirements for maintenance are interesting since they can allocate most of the energy to production traits. This direction of energy can favor the gestational environment and fetal development. Thus, the identification of feed efficient beef cows that have lower maintenance requirements and that produce calves with a good growth potential is a promising approach for beef cattle herds [8], since cows are the animal category that remains in the production system for the longest period of time.

Within this context, the study of the association between RFI and production traits of Nellore cows and the growth potential of their offspring is essential to determine whether lower feed intake, inherent to feed efficiency, has long-term effects on offspring development, body composition and metabolism. It is found in the literature that pregnant beef cows classified as low RFI consumed 1.91 kg DM/day less than high RFI cows for similar levels of body composition (ultrasound-measured fat thickness) and productivity (calf birth weight and calving difficulty) [9].

The hypothesis of the present study is that differences in feed intake, represented by the RFI class of Nellore cows, influence the growth potential, metabolism and gene expression in muscle tissue of their offspring until weaning. The aim of this study was to elucidate the effect of RFI class of Nellore cows on the growth traits and the expression of genes related to lipid metabolism of their offspring in different livestock production environments.

## Material and methods

Two studies were conducted at Centro Avançado de Pesquisa e Desenvolvimento de Bovinos de Corte of the Instituto de Zootecnia (IZ), Secretaria da Agricultura e Abastecimento of the State of São Paulo, in consecutive years. Pregnant Nellore cows of divergent RFI classes raised under different nutritional conditions were studied. After the study, experimental animals returned with no sequels to the production system. The experiments followed the guidelines of animal welfare in accordance with State Law No. 11.977 of São Paulo State, Brazil. All procedures were approved by the Ethics Committee on Animal Use of IZ (protocol number 252–17).

### Cows

Fifty-three contemporaneous Nellore cows belonging to the same line selected for growth and feed efficiency were used. The cows were genotyped using a 77k chip (Neogen, Michigan, USA). The RFI was determined by genomic prediction using relatedness data and missing

genotypes imputed to 700k [10–12]. Based on the RFI thus obtained, the animals were classified as efficient (negative RFI, i.e., animals having RFI below the mean of the herd) or inefficient (positive RFI, i.e., animals having RFI above the mean of the herd).

From the total contemporary group of cows submitted to two fixed-time artificial insemination (FTAI) protocols for two consecutive years, 25 (471 ± 47.4 kg of BW and 749 ± 76 days of age) became pregnant in the first year and 28 became pregnant in the second year (558 ± 43.6 kg of BW and 1107 ± 77 days of age). Semen from the Macegal Nellore bull of IZ, which is classified as negative RFI and is from the same selection line as the cows, was used in both years.

The cows were weighed every 28 days in the morning, without previous fasting, during the second trimester of gestation, totaling three weight recordings. The following ultrasound measures were obtained on the occasion of the last weight recording: 1) rib eye area (REA) and backfat thickness (BFT) measured transversally between the 12$^{th}$ and 13$^{th}$ rib [13]; 2) rump fat thickness (RFT) measured longitudinally over the junction between the *Gluteus medius* and *Biceps femoris* muscles [13]. The images were obtained with a Pie Medical 401347-Aquila apparatus equipped with a 3.5-MHz linear probe (18 cm; Pie Medical Equipment B.V., Maastricht, Netherlands). The images were saved and subsequently analyzed using the Echo Image Viewer 1.0 (Pie Medical Equipment B.V., Maastricht, Netherlands), with a precision of 1 decimal place.

## Calves

Fifty-three Nellore calves were evaluated: 25 calves in Year 1 (13 females and 12 males), including 11 animals born to negative RFI cows and 14 born to positive RFI cows; 28 calves in Year 2 (9 females and 19 males), including 13 animals born to negative RFI cows and 15 born to positive RFI cows.

The calves were weighed at birth and every 28 days until weaning, without previous fasting. Average daily weight gain (ADG) was calculated by regression using all available weights for each animal. The weight at 120 days of age (W120) was calculated from the birth weight and weight gain until 120 days of age. Similarly, the weight at 210 days of age (W210) was calculated from the birth weight and weight gain until 210 days of age. Calves were weaned with 259 ± 38.6 kg of BW and 220 ± 24 days of age in Year 1, and 209 ± 29.1 kg of BW and 254 ± 17 days of age in Year 2. On the occasion of weight recording at weaning, ultrasound measurements were taken at the same anatomical sites as measured in cows (i.e. REA, BFT, and RFT).

## Nutritional management: Year 1

**Prenatal nutrition.** During pregnancy of Year 1, the 25 pregnant cows were kept on *Brachiaria brizantha* cv. Marandu paddock, having 22 ha of area, previously nitrogen fertilized, with mineral supplement in rainy season and protein supplement in dry season. Pasture samples were collected every 28 days during the months corresponding to the second trimester of gestation (April, May, and June) for the determination of nutritional composition. The paddock was sampled at six sites using a quadrant of 1 m$^2$ to obtain a representative sample of the area. The samples collected 5 cm from the ground at each site, were divided into the fractions of leaves, stems, dead material, and intact plant. The nutritional composition of the leaf fraction was used as a proxy of the feed consumed by the cows (Table 1).

The samples were dried in a forced ventilation oven (SOLAB, Piracicaba, SP, Brazil) at 65°C for 72 hours and then ground in a knife mill (Thomas Scientific, Swedesboro, NJ, USA) with a 1-mm sieve. Dry matter (DM) [14; AOAC Official Method 934.01] and mineral matter (MM) [14, AOAC Official Method 942.05] contents were determined. Neutral detergent insoluble fiber was determined using α-amylase without the addition of sodium sulfite [15], and

**Table 1. Chemical composition of the leaf fraction of the pasture available for cows during the second trimester of gestation.**

| Nutritional components | |
|---|---|
| Dry matter (%) | 52.5 |
| Organic matter (%DM) | 97.9 |
| Mineral matter (%DM) | 2.07 |
| Crude protein (%DM) | 5.21 |
| Neutral detergent fiber (%DM) | 73.3 |
| Acid detergent fiber (%DM) | 40.2 |
| Lignin (%DM) | 5.26 |
| Total digestible nutrients (%) | 36.2 |

corrected for ash and protein [16]. Acid detergent insoluble fiber was also determined [17]. Crude protein was analyzed by the combustion method of Dumas for the detection of nitrogen [14, AOAC Official Method 993.13]. Lignin was determined by solubilization of cellulose with sulfuric acid [18]. Total digestible nutrients were estimated from the results of bromatological analysis using the equation of Weiss [19].

Forage availability was calculated as the amount of forageavailable within 1 m$^2$, measured in six points over the area, multiplied by 10,000 m$^2$. The forage available for cows during the second trimester was 10.2 ton of DM/ha, with a leaf-to-stem ratio of 1.78. The proportions of leaf, stem and dead material were 42.6%, 31.8% and 25.6%, respectively.

**Postnatal nutrition.** The cow-calf pairs were kept under feedlot conditions in the same pen from calving to weaning. The diet was formulated to meet the requirements of lactating and pregnant females (RLM 3.2, ESALQ, Piracicaba, SP, Brazil). The animals (cows and calves) were housed in two paired collective paddocks (1,758 m$^2$ and 1,929 m$^2$) with artificial shade (122 m$^2$). In each paddock there was a collective drinker (1,500 L) and five electronic feeders (GrowSafe System$^®$, Vytelle, Kansas City, Missouri, USA) equipped with vertical and horizontal protection bars, which allowed access to only one animal at a time. The animals received an electronic identification tag to recognize each individual in the electronic feeder, which permits daily recording of individual feed intake.

Feed was offered twice a day, being the amount adjusted daily to maintain leftovers at about 10%. The leftovers were removed three times per week and the diet ingredients were sampled weekly for the determination of nutritional composition. The methods of sample preparation and determination of the nutritional components of the diet were the same as those described for leaf fraction (Table 2).

Dry matter intake was calculated as individual feed intake multiplied by the dietary DM content, considering the first 102 days of lactation for cows and the period between 35 days of age and weaning for calves.

## Nutritional management: Year 2

**Prenatal nutrition.** The gestation period of Year 2 coincided with the lactation period of Year 1. Thus, during the second trimester of pregnancy, cow-calf pairs of Year 2 remained under feedlot conditions, with the diet and nutritional management being the same as those described for postnatal nutrition of calves from Year 1.

**Postnatal nutrition.** The 28 cow-calf pairs were kept on *Brachiaria brizantha* cv. Marandu paddock (22 ha of area, previously nitrogen fertilized) from birth to weaning with mineral supplement in rainy season and protein supplement in dry season. As described for the

**Table 2. Nutritional composition of the diet offered to lactating cows and calves.**

| Ingredients | |
|---|---|
| Corn silage (%DM) | 90.34 |
| Soybean meal (%DM) | 8.51 |
| Mineral salt (%DM) | 0.83 |
| Urea (%MS) | 0.32 |
| **Nutritional components** | |
| Dry matter (%) | 41.4 |
| Mineral matter (%DM) | 3.67 |
| Crude protein (%DM) | 11.1 |
| Neutral detergent fiber (%DM) | 51.3 |
| Acid detergent fiber (%DM) | 20.5 |
| Lignin (%DM) | 5.59 |
| Total digestible nutrients (%) | 64.4 |
| Metabolizable energy (Mcal/kg) | 2.42 |

pre-natal management of Year 1, the paddock was sampled at six sites using a quadrant of 1 m$^2$ and the nutritional composition of the leaf fraction was analyzed as a proxy of the feed consumed by cows (Table 3).

Forage availability was calculated as described previously and was 5.31 ton of DM/ha. The leaf-to-stem ratio was 0.73. The proportions of leaf, stem and dead material were 28.4%, 40.6% and 31.0%, respectively.

## Weaning efficiency

The weaning efficiency (WE) was calculated based on calf weight at 205 days (W205) and cow weight at weaning using the equation: PE = (W205/cow weight at weaning)*100. The W205 was calculated as follows: W205 = WW + [(WW+BW)/WA]*(205-WA), where WW is the calf's weight at weaning, BW is the birth weight, and WA is the calf's age in days at weaning [20].

## Muscle biopsy

*Longissimus* muscle biopsies were obtained on the body right side between the 12$^{th}$ and 13$^{th}$ rib of calves at 62 ± 24 days of age in Year 1 and at 56 ± 17 days of age in Year 2. For the biopsy, the lumbar region was shaved, the skin was cleaned and disinfected with degerming

**Table 3. Chemical composition of the leaf fraction of the pasture available for cows and calves during the lactation period.**

| Nutritional components | |
|---|---|
| Dry matter (%) | 47.8 |
| Organic matter (%DM) | 91.2 |
| Mineral matter (%DM) | 8.84 |
| Crude protein (%DM) | 9.13 |
| Neutral detergent fiber (%DM) | 69.5 |
| Acid detergent fiber (%DM) | 34.5 |
| Lignin (%DM) | 4.64 |
| Total digestible nutrients (%) | 51.0 |

chlorhexidine and iodine, and 8 mL of 2% lidocaine hydrochloride without epinephrine was applied (Lidovet, Laboratório Bravet Ltda., Rio de Janeiro, RJ, Brazil) as local anesthetic. Approximately 1-cm long incision was made with a scalpel and a sterilized Bergstrom cannula (AgnThos, Eskildstuna, Södermanland, Sweden) was used to obtain 1 to 2 g of muscle tissue. After the procedure, the incision was washed with sterile saline/water, treated with iodine and antibiotic spray, and sutured with veterinary tissue adhesive. The calves were monitored until healing and the medication was applied again if necessary. The muscle tissue samples were immediately transferred to an identified cryogenic tube and frozen in liquid nitrogen for subsequent analysis of gene expression.

Close to the time of weaning, when the animals of Year 1 were 214 ± 24 days of age and the animals of Year 2 were 247 ± 17 days of age, a biopsy was performed to collect *Longissimus* muscle samples from body left side between the 12th and 13th rib for histological and gene expression analysis. For this procedure, in addition to the 8 mL of intradermally applied anesthetic, the calves received an additional 12 mL applied intramuscularly. A 10-cm skin incision was made with a scalpel in the cranio-caudal direction parallel to the dorsal midline of the animal. After the incision, the adipose tissue and epimysium were dissected for exposure of the muscle and two incisions were made parallel to the midline of the animal, removing a fragment that measured approximately 1 cm in width x 2 cm in length x 0.5 cm in depth for histological analysis.

Once the tissue sample was obtained, the muscle was immediately sutured with absorbable polyglycolic acid sutures (No.2, Atramat®, Ciudad de México, DF, Mexico) and the skin was sutured with 60-mm nylon suture. Iodine and antibiotic spray were applied to the site of the incision for healing. The calves received an injectable anti-inflammatory agent and antibiotic and were monitored for 14 days after the procedure. The medications were applied again, if necessary, until removal of the external sutures.

## Histological analysis

The muscle samples were fixed in 10% formalin for 48 hours. After fixation, the tissues were dehydrated in an increasing alcohol series (80, 85, 90, 95 and 100% ethanol) and submitted to two clearing sequences with xylene. The material was embedded in paraffin blocks and 5-μm sections were cut with a rotary microtome. Three slides were prepared per tissue sample per animal, totaling six sections. The tissue sections were stained with hematoxylin and eosin [21]. The samples were analyzed under an Olympus CX31 light microscope coupled to an Olympus SC30 camera (Olympus Corp., Tokyo, Honshu Island, Japan) for image capture using a 40X objective. After acquisition of the images, 50 fibers per sample were analyzed using the ImageJ® software (National Institutes of Health, Bethesda, Maryland, USA). The Straight tool of the software was used to measure the diameter and area in μm.

## Gene expression

The primers of the target and reference genes were selected based on the literature [22] using registered sequences published in the GenBank database of the National Center for Biotechnology Information (NCBI) platform (Table 4).

Total RNA was extracted from the muscle samples using QIAzol (Qiagen, Valencia, CA, USA) and treated with DNA-free DNase (Ambion, Austin, TX, USA) according to manufacturer recommendations. For analysis of the rRNA bands (28S and 18S), total RNA was submitted to electrophoresis on 1% agarose gel stained with GelRed (Biotium, Hayward, CA, USA) and visualized under a UVItec FireReader XS D-77Ls-20M camera (UVItec, Cambridge, UK). The cDNA was synthesized using the cDNA Reverse Transcription Kit (Applied Biosystems,

**Table 4. Sequence (5' to 3') and efficiency of the primers used for qRT-PCR.**

| Symbol | Gene | Forward (F) and reverse (R) | Accession number | Amplicon (bp) | $R^2$ | Efficiency |
|---|---|---|---|---|---|---|
| PPARA | Peroxisome proliferator-activated receptor α | F CAATGGAGATGGTGGACACA<br>R TTGTAGGAAGTCTGCCGAGAG | NM_001034036.1 | 95 | 0.992 | 99.2 |
| PPARG | Peroxisome proliferator-activated receptor gamma | F GCAATCAAAGTGGAGCCTGT<br>R CCATGAGGGAGTTGGAAGG | NM_181024.2 | 94 | 0.973 | 100 |
| SREBF1 | Sterol regulatory element-binding protein-1c | F GAGCCACCCTTCAACGAA<br>R TGTCTTCTATGTCGGTCAGCA | NM_001113302.1 | 88 | 0.985 | 94.6 |
| LPL | Lipoprotein lipase | F CTCAGGACTCCCGAAGACAC<br>R GTTTTGCTGCTGTGGTTGAA | NM_001075120.1 | 98 | 0.99 | 96.7 |
| FABP4 | Fatty acid binding protein 4 | F GGATGGAAAATCAACCACCA<br>R GTGGCAGTGACACCATTCAT | NM_174314.2 | 84 | 0.991 | 99 |
| ACACA | Acetyl CoA carboxylase alpha | F TGAAGAAGCAATGGATGAACC<br>R TTCAGACACGGAGCCAATAA | NM_174224.2 | 88 | 0.994 | 96.6 |
| FASN | Fatty acid synthase | F ATCAACTCTGAGGGGCTGAA<br>R CAACAAAACTGGTGCTCACG | U34794.1 | 83 | 0.974 | 99.5 |
| SCD1 | Stearoyl-CoA desaturase | F ACCATCACAGCACCTCCTTC<br>R ATTTCAGGGCGGATGTCTTC | NM_173959.4 | 95 | 0.991 | 98 |
| ACOX | Acyl-coenzyme A oxidase 1 | F GCTGTCCTAAGGCGTTTGTG<br>R ATGATGCTCCCCTGAAGAAA | BC102761.2 | 83 | 0.994 | 99 |
| CPT2 | Carnitine palmitoyl transferase 2 | F CTATTCCCAAACTTGAAGAC<br>R TTTTCCTGAACTGGCTGTCA | NM_001045889.2 | 81 | 0.952 | 98 |
| β-actin | β-actin | F GTCCACCTTCCAGCAGATGT<br>R CAGTCCGCCTAGAAGCATTT | NM_173979.3 | 90 | 0.996 | 105 |
| CASC3 | Cancer susceptibility candidate 3 | F GGACCTCCACCTCAGTTCAA<br>R GTCTTTGCCGTTGTGATGAA | NM_001098069.1 | 85 | 0.976 | 98 |

Foster City, CA, USA) and the samples were stored at -20˚C. qRT-PCR was carried out in an Eppendorf Realplex Real-Time PCR System (Eppendorf, Hamburg, Germany) using the SYBR Green detection system (Applied Biosystems, Foster City, CA, USA) and the cDNA of the samples [23].

The efficiency of the primers was determined by constructing standard curves for the genes studied using the following dilutions: 1:5, 1:25, 1:125, 1:625, and 1:3,125. The results were normalized using the cycle threshold (CT) obtained from the expression of the reference genes *β-actin* and cancer susceptibility candidate 3 (*CASC3*). Reference genes were chosen according to literature guidelines [22]. The relative expression levels were calculated [24] based on the CT values that were corrected for the amplification efficiency of each primer.

## Statistical analysis

The data of each year were analyzed in a completely randomized design. Analysis of variance was performed using the PROC MIXED procedure of the SAS program (SAS Institute Inc., Cary, NC, USA), considering RFI class and FTAI protocol as the fixed effects for the variables measured in cows. For the variables measured in calves, RFI class, FTAI protocol and sex were included as fixed effects. The Shapiro-Wilk test was applied to evaluate the normality of the collected data. Since there was no normal distribution, the data were transformed using the PROC RANK procedure of SAS (SAS Institute Inc., Cary, NC). The interactions between effects were tested and removed from the model because they were not significant. Means were calculated by the least squares method and compared by the *t*-test at a probability level of 5%. Tendencies were defined when $0.05 < P \leq 0.10$ for the performance traits and when $0.05 < P \leq 0.15$ for the gene expression data.

## Results

Considering both years, the mean RFI of cows was -0.087 kg DM/day. This value is within the range of the trait for a herd under selection for feed efficiency (Table 5).

In both years, the average BW of cows increased during the second trimester of gestation, even when fetal weight was deducted, demonstrating energy availability for weight gain in both nutritional situations. In Year 1, the cows weighed on average 501 ± 50.4, 516 ± 51.5 and 527 ± 53.4 kg in the first, second and third month of the second trimester of gestation, respectively, with an ADG of 0.381 ± 0.178 kg/days. In Year 2, the average BW of cows was 594 ± 46.4, 611 ± 46.8 and 626 ± 46.9 kg in the first, second and third month of the second trimester of gestation, respectively, with an ADG of 0.501 ± 0.241 kg/days. The higher weights during pregnancy observed in Year 2 were expected since gestation occurred under feedlot conditions, which are characterized by greater nutrient availability of the diet compared to tropical pastures.

In Year 1, the cows gestated under the typical nutritional conditions of pastures and nursed under better nutritional conditions, with feed being offered in a trough. The animals reached an average BW of 615 ± 57.9 kg at weaning, corresponding to a weight gain of 88 kg between the end of the second trimester of gestation and weaning of the calves. In Year 2, the cows gestated under feedlot conditions and nursed their calves under grazing conditions. The average BW at weaning was 523 ± 37.8 kg, corresponding to a weight loss of 103 kg between the end of the second trimester of gestation and weaning of the calves.

The average birth weights of the calves were similar in the two years (33.6 ± 4.58 kg in Year 1 and 33.5 ± 6.32 kg in Year 2), showing that the difference in the nutritional condition to which the cows were exposed during pregnancy did not affect fetal growth. However, the same difference in nutritional condition to which the cow-calf pair was exposed had an expressive effect on the calf's growth from birth to weaning, as demonstrated by the average BW at 120 and 210 days of age (155 vs 118 kg for W120 and 251 vs 184 kg for W210 in Years 1 and 2, respectively). These differences were also reflected in the body composition of the calves, with the following mean values for REA, BFT and RFT, respectively: 46.0 ± 6.11 cm$^2$, 2.26 ± 0.888 mm and 5.72 ± 1.43 mm in Year 1 and 34.8 ± 6.48 cm$^2$, 1.26 ± 0.552 mm and 2.39 ± 0.641 mm in Year 2. The weaning efficiency of calves was 40.9 ± 5.92% in Year 1 and 33.1 ± 4.87% in Year 2, corresponding to a 19.1% greater efficiency of calves of Year 1 compared to those of Year 2.

No significant differences in the performance traits of cows during the second trimester of gestation or postnatal growth of calves were found between RFI classes in Year 1 (Table 6). These results show that the efficient class of the cow was not associated with calf performance from birth to weaning under the environmental conditions of this year.

Despite the similarity in the performance traits of cows during pregnancy and in the postnatal growth traits of calves, differences in the expression of genes involved in lipid metabolism were detected between RFI classes at birth (Fig 1A). Offspring of negative RFI cows tended to have lower expression of the sterol regulatory element-binding protein 1c (SREBF1) gene and consequently lower expression of the lipoprotein lipase (LPL) and fatty acid binding protein 4 (FABP4) genes. However, there was no difference between cow RFI classes for the peroxisome proliferator-activated receptor (PPARA and PPARG), acetyl CoA carboxylase alpha (ACACA), fatty acid synthase (FASN), stearoyl-CoA desaturase (SCD1), carnitine palmitoyltransferase 2 (CPT2), or acyl-coenzyme A oxidase 1 (ACOX) genes. Despite the effect observed at birth, the expression of all genes studied was similar in both RFI classes of cows close to weaning (Fig 1B).

In Year 2, negative RFI cows had higher BFT and tended to have higher RFT in the second trimester of gestation than positive RFI cows, and their calves tended to be heavier at birth and

**Table 5. Descriptive statistics of the variables studied in Nellore cows and calves of Years 1 and 2.**

| Variable | n | Mean | SD | CV | Maximum | Minimum |
|---|---|---|---|---|---|---|
| **Year 1** | | | | | | |
| **Cows** | | | | | | |
| RFI, kg/day | 25 | -0.086 | 1.49 | - | 2.50 | -4.85 |
| W1, kg | 25 | 501 | 50.4 | 10.1 | 586 | 412 |
| W2, kg | 25 | 516 | 51.5 | 9.98 | 608 | 422 |
| W3, kg | 25 | 527 | 53.4 | 10.1 | 621 | 422 |
| WW, kg | 25 | 615 | 57.9 | 9.41 | 740 | 492 |
| ADG, kg/day | 25 | 0.381 | 0.178 | 46.8 | 0.719 | 0.098 |
| REA, cm$^2$ | 25 | 79.4 | 5.21 | 6.56 | 87.7 | 64.3 |
| BFT, mm | 25 | 5.69 | 1.73 | 30.3 | 8.94 | 2.83 |
| RFT, mm | 25 | 7.31 | 2.17 | 29.7 | 12.3 | 3.07 |
| DMI, kg/day | 25 | 12.0 | 1.69 | 14.0 | 14.5 | 9.0 |
| **Calves** | | | | | | |
| BW, kg | 25 | 33.6 | 4.58 | 13.6 | 45.0 | 26.0 |
| W120, kg | 24 | 155 | 18.1 | 11.7 | 200 | 116 |
| W210, kg | 24 | 251 | 28.2 | 11.2 | 308 | 188 |
| ADG, kg/day | 24 | 1.03 | 0.121 | 11.7 | 1.26 | 0.772 |
| REA, cm$^2$ | 24 | 46.0 | 6.11 | 13.3 | 59.1 | 38.0 |
| BFT, mm | 24 | 2.26 | 0.888 | 39.3 | 3.80 | 0.000 |
| RFT, mm | 24 | 5.72 | 1.43 | 25.0 | 8.60 | 3.40 |
| DMI, kg/day | 24 | 2.64 | 0.895 | 33.9 | 4.42 | 1.51 |
| PE, % | 24 | 40.9 | 5.92 | 14.5 | 59.5 | 29.6 |
| **Year 2** | | | | | | |
| **Cows** | | | | | | |
| RFI, kg/day | 28 | -0.088 | 0.681 | - | 1.28 | -1.34 |
| W1, kg | 28 | 594 | 46.4 | 7.81 | 711 | 476 |
| W2, kg | 28 | 611 | 46.8 | 7.66 | 727 | 498 |
| W3, kg | 28 | 626 | 46.9 | 7.51 | 739 | 527 |
| WW, kg | 28 | 523 | 37.8 | 7.22 | 591 | 471 |
| ADG, kg/day | 28 | 0.501 | 0.241 | 48.1 | 1.03 | 0.147 |
| REA, cm$^2$ | 28 | 81.5 | 5.20 | 6.38 | 93.3 | 73.4 |
| BFT, mm | 28 | 8.50 | 1.58 | 18.6 | 12.2 | 5.73 |
| RFT, mm | 28 | 10.4 | 1.62 | 15.5 | 15.8 | 7.87 |
| **Calves** | | | | | | |
| BW, kg | 28 | 33.5 | 6.32 | 18.9 | 46.0 | 20.0 |
| W120, kg | 28 | 118 | 14.1 | 12.0 | 144 | 85.2 |
| W210, kg | 27 | 184 | 23.9 | 13.0 | 223 | 137 |
| ADG, kg/day | 27 | 0.761 | 0.130 | 17.1 | 1.07 | 0.537 |
| REA, cm$^2$ | 27 | 34.8 | 6.49 | 18.7 | 45.3 | 21.3 |
| BFT, mm | 27 | 1.26 | 0.552 | 43.7 | 2.30 | 0.000 |
| RFT, mm | 27 | 2.39 | 0.640 | 26.7 | 3.80 | 1.10 |
| Area, μm$^2$ | 27 | 3074 | 573 | 18.6 | 3875 | 1221 |
| Diameter, μm | 27 | 60.9 | 6.81 | 11.2 | 71.0 | 39.5 |
| WE, % | 27 | 33.1 | 4.87 | 14.7 | 40.8 | 23.6 |

n: Number of observations, SD: Standard deviation, CV: Coefficient of variation, RFI: Residual feed intake, W1: Weight in the 1st month of the second trimester of gestation, W2: Weight in the 2nd month of the second trimester of gestation, W3: Weight in the 3rd month of the second trimester of gestation, WW: Weight at weaning of the calves, BW: Birth weight, W120: Weight at 120 days of age, W210: Weight at 210 days of age, ADG: Average daily gain, REA: Rib eye area, BFT: Backfat thickness, RFT: Rump fat thickness, DMI: Dry matter intake, WE: Weaning efficiency, Area: Muscle fiber area, Diameter: Muscle fiber diameter.

**Table 6. Performance of Nellore cows of different residual feed intake (RFI) classes during the second trimester of gestation and postnatal growth of their calves in Year 1.**

| | Negative RFI | Positive RFI | P-value |
|---|---|---|---|
| **Cows** | | | |
| RFI | -0.184 ± 0.467 | 0.376 ± 0.417 | 0.071[T] |
| W1, kg | 487 ± 15.3 | 505 ± 13.6 | 0.363[NS] |
| W2, kg | 501 ± 15.4 | 520 ± 13.7 | 0.365 [NS] |
| W3, kg | 508 ± 15.6 | 534 ± 14.0 | 0.219 [NS] |
| WW, kg | 604 ± 16.6 | 610 ± 14.8 | 0.806 [NS] |
| ADG, kg/day | 0.301 ± 0.048 | 0.402 ± 0.043 | 0.124 [NS] |
| REA, cm$^2$ | 81.0 ± 1.43 | 79.2 ± 1.27 | 0.296 [NS] |
| BFT, mm | 5.48 ± 0.542 | 5.97 ± 0.484 | 0.496 [NS] |
| RFT, mm | 7.23 ± 0.681 | 7.59 ± 0.608 | 0.698 [NS] |
| DMI, kg/day | 11.8 ± 0.501 | 11.9 ± 0.447 | 0.974 [NS] |
| **Calves** | | | |
| BW, kg | 35.0 ± 1.17 | 33.1 ± 1.05 | 0.226 [NS] |
| W120, kg | 159 ± 5.05 | 152 ± 4.64 | 0.317 [NS] |
| W210, kg | 253 ± 8.20 | 247 ± 7.54 | 0.627 [NS] |
| ADG, kg/day | 1.04 ± 0.036 | 1.02 ± 0.033 | 0.737 [NS] |
| REA, cm$^2$ | 45.7 ± 1.52 | 44.4 ± 1.40 | 0.520 [NS] |
| BFT, mm | 2.17 ± 0.274 | 2.21 ± 0.252 | 0.906 [NS] |
| RFT, mm | 5.74 ± 0.388 | 5.43 ± 0.357 | 0.574 [NS] |
| DMI, kg/day | 2.68 ± 0.290 | 2.60 ± 0.267 | 0.844 [NS] |
| WE, % | 41.3 ± 1.67 | 41.7 ± 1.54 | 0.867 [NS] |

W1: Weight in the 1$^{st}$ month of the second trimester of gestation, W2: Weight in the 2$^{nd}$ month of the second trimester of gestation, W3: Weight in the 3$^{rd}$ month of the second trimester of gestation, WW: Weight at weaning of the calves, BW: Birth weight, W120: Weight at 120 days of age, W210: Weight at 210 days of age, ADG: Average daily gain, REA: Rib eye area, BFT: Backfat thickness, RFT: Rump fat thickness, DMI: Dry matter intake, WE: Weaning efficiency.

[*]: <0.05;

[T]: 0.05<[T]<0.10;

[NS]: Not significant.

to have less rump fat at weaning (Table 7). No significant differences between maternal RFI classes were found for the other variables studied.

In Year 2, the expression of genes involved in lipid metabolism, were similar between maternal RFI classes for transcription factors (*SREBF1*, *PPARA* and *PPARG*), uptake genes (*LPL* and *FABP4*), lipogenic genes (*ACACA*, *FASN* and *SCD1*), and lipolytic genes (*CPT2* and *ACOX*) at birth or weaning (Fig 2).

## Discussion

Feed efficiency is relevant for beef cattle production systems since feed costs account for the highest percentage of total production costs. In beef cattle herds, the cow category remains in the production system for the longest period of time and requires the largest supply of inputs; thus, small daily variations in the feed intake of this category would have a more expressive effect on the efficiency of the whole system [25]. The RFI is a measure of feed efficiency independent of the BW and growth rate of the animals [6]. The measurement of RFI in cows and the selection of efficient animals are an alternative to reduce inputs necessary for beef production, which would have economic and environmental benefits.

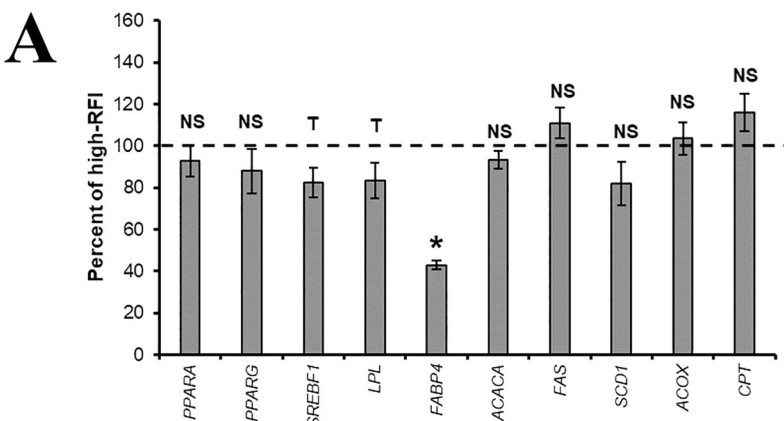

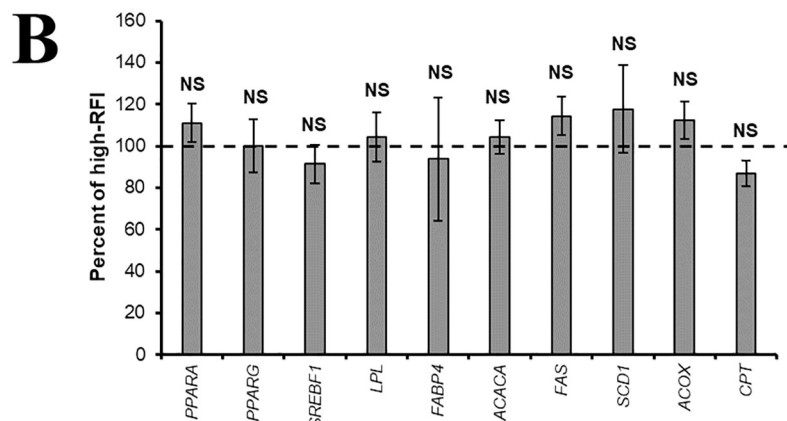

**Fig 1. Gene expression in *Longissimus* muscle of Nellore calves born to cows classified as positive (dotted line) and negative (columns) residual feed intake (RFI) in Year 1.** (A) Birth, (B) Weaning. The bars represent the standard deviation of the mean. Values with an asterisk (*) differ significantly (P ≤ 0.05) and T indicates a tendency (0.05 < P ≤ 0.15). NS: Not significant (P > 0.15).

The difference in dry matter intake between RFI classes was 0.560 kg/day in Year 1 (-0.184 kg/day for negative RFI and +0.376 kg/day for positive RFI) and 0.358 kg/day in Year 2 (-0.280 kg/day for negative RFI and +0.078 kg/day for positive RFI). Studies of RFI in beef cows in the literature reported an average dry matter intake of 9.6 kg/day for animals classified as low RFI, of 11.2 kg/day for those classified as medium RFI, and of 12.6 kg/day for those classified as high RFI [26] or of 12.0 kg/day for negative RFI cows and of 13.3 kg/day for positive RFI cows [27]. These results indicate expressive differences in feed intake between feed efficiency classes of cows.

The main function of cows in beef cattle herds is the production of healthy calves with a good growth potential. Thus, studying the effects of the choice of feed-efficient cows based on RFI on the calves produced is of fundamental importance in order to avoid losses in production traits of the herds. It is equally important to know the effect of the pre- and postnatal environments on the growth traits of calves. These are the reasons why the results of Years 1 and 2 were reported sequentially.

The performance of pasture-raised animals is directly related to the availability and quality of forage, which interferes with feed intake and consequently with nutrient intake [28]. Years 1

**Table 7. Performance of Nellore cows of different residual feed intake (RFI) classes during the second trimester of gestation and postnatal growth of their calves in Year 2.**

| | Negative RFI | Positive RFI | *P*-value |
|---|---|---|---|
| **Cows** | | | |
| RFI, kg/d | -0.280 ± 0.202 | 0.078 ± 0.235 | 0.066[T] |
| W1, kg | 572 ± 12.8 | 590 ± 14.9 | 0.301 [NS] |
| W2, kg | 594 ± 13.5 | 607 ± 15.7 | 0.477 [NS] |
| W3, kg | 611 ± 13.8 | 623 ± 16.1 | 0.554 [NS] |
| WW, kg | 517 ± 11.2 | 511 ± 13.3 | 0.709 [NS] |
| ADG, kg/day | 0.626 ± 0.066 | 0.512 ± 0.076 | 0.211 [NS] |
| REA, cm$^2$ | 82.6 ± 1.51 | 79.3 ± 1.76 | 0.123 [NS] |
| BFT, mm | 8.72 ± 0.433 | 7.45 ± 0.503 | 0.040 * |
| RFT, mm | 10.6 ± 0.451 | 9.42 ± 0.524 | 0.076[T] |
| **Calves** | | | |
| BW, kg | 34.3 ± 1.90 | 29.8 ± 2.08 | 0.068[T] |
| W120, kg | 118 ± 4.84 | 116 ± 5.30 | 0.738 [NS] |
| W210, kg | 184 ± 7.91 | 186 ± 8.69 | 0.827 [NS] |
| ADG, kg/day | 0.810 ± 0.034 | 0.811 ± 0.037 | 0.971 [NS] |
| REA, cm$^2$ | 33.1 ± 2.23 | 34.1 ± 2.44 | 0.739 [NS] |
| BFT, mm | 1.08 ± 0.186 | 1.30 ± 0.204 | 0.359 [NS] |
| RFT, mm | 2.23 ± 0.203 | 2.66 ± 0.222 | 0.097[T] |
| Area, μm$^2$ | 2977 ± 196 | 3060 ± 215 | 0.734 [NS] |
| Diameter, μm | 59.8 ± 2.34 | 60.9 ± 2.57 | 0.693 [NS] |
| WE, % | 34.0 ± 1.51 | 33.8 ± 1.72 | 0.928 [NS] |

W1: Weight in the 1$^{st}$ month of the second trimester of gestation, W2: Weight in the 2$^{nd}$ month of the second trimester of gestation, W3: Weight in the 3$^{rd}$ month of the second trimester of gestation, WW: Weight at weaning of the calves, BW: Birth weight, W120: Weight at 120 days of age, W210: Weight at 210 days of age, ADG: Average daily gain, REA: Rib eye area, BFT: Backfat thickness, RFT: Rump fat thickness, Area: Muscle fiber area, Diameter: Muscle fiber diameter, WE: Weaning efficiency.

*: <0,05;

T: 0,05<T<0,10;

NS: Not significant.

and 2 used feeds with highly different nutritional compositions (pasture and diet offered in a trough), with the forage quality being below satisfactory standards during some periods.

As expected, a difference of 19% was found in the comparison of calf production efficiency between Years 1 and 2, which is mostly explained by the variation in nutritional conditions and physiological differences between studied years. Thus, the adoption of feed efficiency measures in order to improve productive efficiency must be considered in systems that meet the basic requirements for animal nutrition.

The ADG of cows during the second trimester of gestation was 0.301 and 0.402 kg/day in Year 1 and 0.626 and 0.512 kg/day in Year 2 for negative and positive RFI cows, respectively. The difference between years was mainly due to the nutritional condition established. ADG under feedlot conditions of 0.81, 0.84 and 0.82 kg/day for low, medium and high RFI cows, respectively, were reported in the literature [26]. The ADG of cows during the second trimester found in Years 1 and 2 showed that pregnant cows met the nutritional requirements, even in the situation of low forage quality (pregnancy studied in Year 1). The cows did not enter a negative energy balance during gestation, a key factor for the formation of muscle tissue in the fetus which can affect its growth potential during different phases of life.

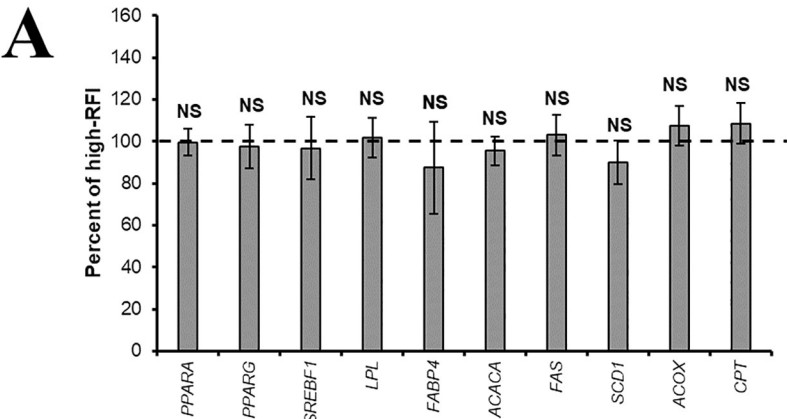

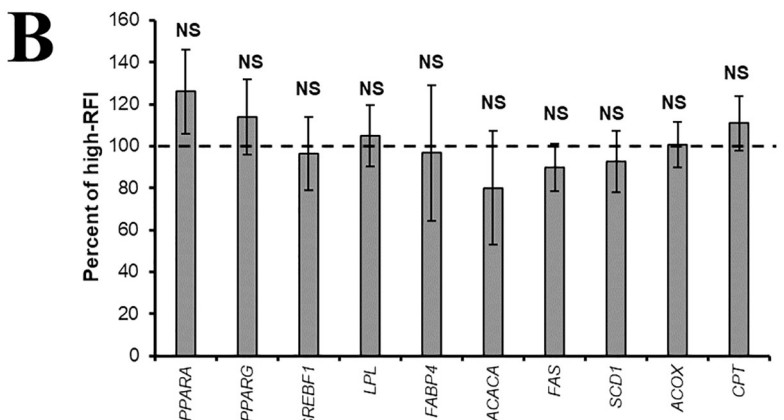

**Fig 2. Gene expression in *Longissimus* muscle of Nellore calves born to cows classified as positive (dotted line) and negative (columns) residual feed intake (RFI) in Year 2.** (A) Birth, (B) Weaning. The bars represent the standard deviation of the mean. Values with an asterisk (*) differ significantly ($P \leq 0.05$) and T indicates a tendency ($0.05 < P \leq 0.15$). NS: Not significant ($P > 0.15$).

In conventional beef cattle production systems, the suckling period of calves born in one year coincides with the gestation period of calves of the following year. The calves of Year 1, which were gestated under grazing conditions, spent the suckling period under feedlot conditions together with their mothers that were pregnant with the calves of Year 2. Thus, due to the better nutritional conditions, these animals had greater ADG during the period from birth to weaning than calves of Year 2, regardless of maternal RFI class. Calves born to cows that received protein supplementation during pregnancy had higher weights at birth, 4 months of age and weaning than calves born to not supplemented cows [29].

The higher BW of calves of Year 1 compared to Year 2 resulted in greater REA, BFT and RFT. The ultrasound measures of calves born to non-supplemented and supplemented cows were compared and significant differences were detected between treatments: 49.6 vs 54.9 cm$^2$ for REA and 5.6 vs 6.6 mm for BFT, respectively [30].

Weaning efficiency was calculated in the two nutritional situations evaluated (Years 1 and 2). Negative and positive RFI cows produced calves with the same efficiency, which is an advantage for negative RFI cows that consumed less feed for the same weaning efficiency. The

weaning efficiency of Nellore cow-calf pairs of different RFI classes were reported in the literature, being similar for negative and positive RFI animals (40.6 vs 40.0%) [27].

It is known that cows have increased amino acid requirements during fetal development and that both the lack and excess of these nutrients can alter the metabolic pathways of the fetus. It is reported in the literature that nutritional restriction of the pregnant cow can affect the expression of the *PPARG* gene in adipose tissues [31]. The *PPARG* gene uses fatty acids as endogenous ligands and the expression of the gene is regulated nutritionally. Thus, amino acids present in maternal milk can affect the expression of *PPARG* in calves [32]. The *FABP4* gene is expressed in adipocytes and influences the coding of proteins related to fatty acid metabolism necessary for fat deposition [33]. As the expression of the *FABP4* gene occurs, intramuscular fat develops [34]. In ruminants, 55 to 60% of free fatty acids are derived from the hydrolysis of triglycerides mediated by the *LPL* gene [35]. This gene plays a key role in lipid uptake and in the formation of fat cells and also regulates the BW and energy balance of animals by controlling triglycerides in adipose and muscle tissues [36,37].

Three factors interfere with the lipid balance: diet, biosynthesis, and catabolism through β-oxidation [38]. Changes in synthesis and degradation of lipid balance are related to an increase or decrease of intramuscular fat [39]. The *SCD1* gene is involved in lipid metabolism and BW control [40]. In cattle, *SCD1* is associated with marbling and milk production of females [41,42]. The protein *FASN* is mainly involved in fatty acid synthesis and lipid metabolism [43] and its levels are controlled by the mRNA transcription rate [44]. Beta-oxidation of fatty acids is initiated by *ACOX1*. Like *SDC1* and *FASN*, this enzyme is significant for lipid metabolism.

In the present study, expression of the *FABP4* gene at birth was lower in calves born to negative RFI cows; however, when evaluated close to weaning, the expression of this gene was similar in calves born to negative and positive RFI cows. In addition, there was a tendency towards lower expression of the *SREBF1* and *LPL* genes in calves born to negative RFI cows immediately after birth. All of these genes are associated with the formation of fat cells, indicating that adipogenesis occurs later during the fetal and neonatal period in calves born to negative RFI cows.

Despite the similarity between maternal RFI classes in terms of the growth traits of their offspring, the gene expression data of Year 1 suggest that RFI exerts an effect on adipogenesis and consequently affects fat deposition in the offspring during the finishing phase. The results of this study indicate that RFI class exerts its effect mainly under conditions of maternal nutritional restriction since no effect on the expression of genes involved in lipid metabolism was observed in Year 2, in which cows gestated under feedlot conditions. In cattle, adipogenesis starts in mid-gestation and lasts throughout the life of the offspring [45]. Although there was no significant difference in the expression of the transcription factor *PPARG*, one of the main markers of adipogenesis, between maternal RFI classes, the lower expression of *SREBF1* at birth in calves born to negative RFI cows suggests reduced adipogenesis in these animals during the fetal and neonatal period. However, during the neonatal period, the adipogenesis of intramuscular fat becomes more effective because of greater differentiation of progenitor cells to the intramuscular adipocyte lineage [45], a fact that may have prevented the effect of the maternal's RFI class from persisting until weaning since the nutritional conditions were more favorable.

The *SREBF1* transcription factor is involved in the terminal events of adipose cell differentiation. In the nucleus, this factor activates the transcription of genes that encode enzymes involved in lipid metabolism such as *LPL* and *FABP4* [46–48]. In Year 1, as observed for *SREBF1*, offspring of negative RFI cows exhibited lower expression of the *LPL* and *FABP4* genes.

The *FABP4* gene is mainly expressed in mature adipocytes. This gene encodes a protein that participates in the absorption and transport of fatty acids, being involved in fat deposition [49]. Thus, the lower expression of *FABP4* may be an indicator of a reduction in the formation of adipocytes in muscle tissue during the fetal period, which may affect fat deposition during the finishing phase of these animals.

## Conclusion

The pre- and postnatal nutritional conditions influence the deposition of body tissues at the beginning of life in Nellore cattle. Nellore cows classified as negative RFI are beneficial for global livestock production systems since they consume less feed and produce calves with a growth potential similar to those produced by positive RFI cows. Adipogenesis during the fetal period is reduced in calves born to negative RFI cows that gestated under the nutritional conditions of tropical pastures.

## Supporting information

**S1 Fig. Genetic value for RFI from the entire Nellore progeny group born in 2014 at Instituto de Zootenia.**
(DOCX)

**S2 Fig. Experimental cow-calf pairs performance data.**
(DOCX)

## Author Contributions

**Conceptualization:** Maria Eugênia Zerlotti Mercadante, Sarah Figueiredo Martins Bonilha.

**Data curation:** Maria Eugênia Zerlotti Mercadante, Sarah Figueiredo Martins Bonilha.

**Formal analysis:** Márcio Machado Ladeira, Priscilla Dutra Teixeira, Camila Delveaux Araujo Batalha, Maria Eugênia Zerlotti Mercadante, Sarah Figueiredo Martins Bonilha.

**Funding acquisition:** Sarah Figueiredo Martins Bonilha.

**Investigation:** Ana Carolina Almeida Rollo de Paz, Márcio Machado Ladeira, Roberta Carrilho Canesin, Camila Delveaux Araujo Batalha, Sarah Figueiredo Martins Bonilha.

**Methodology:** Márcio Machado Ladeira, Priscilla Dutra Teixeira, Roberta Carrilho Canesin, Camila Delveaux Araujo Batalha, Sarah Figueiredo Martins Bonilha.

**Project administration:** Roberta Carrilho Canesin, Sarah Figueiredo Martins Bonilha.

**Resources:** Sarah Figueiredo Martins Bonilha.

**Supervision:** Márcio Machado Ladeira, Roberta Carrilho Canesin, Maria Eugênia Zerlotti Mercadante, Sarah Figueiredo Martins Bonilha.

**Validation:** Márcio Machado Ladeira, Priscilla Dutra Teixeira, Camila Delveaux Araujo Batalha, Sarah Figueiredo Martins Bonilha.

**Visualization:** Sarah Figueiredo Martins Bonilha.

**Writing – original draft:** Ana Carolina Almeida Rollo de Paz, Camila Delveaux Araujo Batalha.

**Writing – review & editing:** Márcio Machado Ladeira, Sarah Figueiredo Martins Bonilha.

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
