## [Decision Letter · Decision Letter 0]

7 Feb 2022

PONE-D-21-37544Performance and muscle lipogenesis in offspring of Nellore cows of different residual feed intake classesPLOS ONE

Dear Dr. Bonilha,

Thank you for submitting your manuscript to PLOS ONE. After careful consideration, we feel that it has merit but does not fully meet PLOS ONE’s publication criteria as it currently stands. Therefore, we invite you to submit a revised version of the manuscript that addresses the points raised during the review process.

We look forward to receiving your revised manuscript.

Kind regards,

Marcio Duarte, PhD

Academic Editor

PLOS ONE

Journal Requirements:1. When submitting your revision, we need you to address these additional requirements.

2. In your Methods section, please include a comment about the state of the animals following this research. Were they euthanized or housed for use in further research? If any animals were sacrificed by the authors, please include the method of euthanasia and describe any efforts that were undertaken to reduce animal suffering.

(This work was supported by São Paulo

Research Foundation (FAPESP): grant #2017/06709-2 provided to SFMB; and #2018/20080-2 provided to CDAB.

This work was also supported by Coordination for the Improvement of Higher Education Personnel (CAPES, Finance Code 001), providing grant to ACARP.

The funders had no role in study design, data collection and analysis, decision to publish, or preparation of the manuscript.)

Reviewers' comments:

Reviewer's Responses to Questions

**Comments to the Author**

1. Is the manuscript technically sound, and do the data support the conclusions?

Reviewer #1: Partly

Reviewer #2: Yes

2. Has the statistical analysis been performed appropriately and rigorously? 

Reviewer #1: No

Reviewer #2: Yes

3. Have the authors made all data underlying the findings in their manuscript fully available?

Reviewer #1: Yes

Reviewer #2: Yes

4. Is the manuscript presented in an intelligible fashion and written in standard English?

Reviewer #1: No

Reviewer #2: Yes

5. Review Comments to the Author

Reviewer #1: Overall considerations: This study evaluates the effects of maternal RFI classification on calf gene expression and growth performance. This reviewer believes authors should pay close attention to the statistical analyses, description of methods, and writing style. The manuscript is lacking flow. For example, introduction does not address well the research project and the abstract does not reflect the studies performed. It does not mention that 2 separately studies were conducted. Regarding the statistics, for instance, AI is included in the fixed model when it was not supposed to be since researchers were not testing different protocols. In fact, the AI protocol was the same for every cow, including the sire used. Therefore, no needed to be included in the fixed model. This reviewer thinks that both experiments should be analyzed as a single project, generating the interaction between feeding system x RFI. For the calf data, sex was included in the fixed term, why an interaction was not tested then (maternal RFI|SEX)? P-values for sex or AI were not even reported even though they were in the fixed model. The gene expression data could have been analyzed as a repeated measure, therefore, conclusion could be drawn when comparing the pre-weaning vs. post-weaning values. The introduction section can be improved by stating why testing different feeding systems are needed. It would justify the 2 experiments. Materials and methods are missing important information about animals/pen, days of the experiment and so on. Table 5 is not needed and could be removed from the manuscript. English/written style could be improved. This reviewer suggests authors to improve the writing style, table descriptions, and statistics analyses. Some other detailed information can be found below.

Title: This reviewer suggests changing the title to: “Performance and muscle lipogenesis of calves born to Nellore cows with different residual feed intake classification”.

Introduction does not address why authors were testing 2 systems (grazing and feedlot/dry lot pen). Why was that tested?

Line 16: Replace to “This study aimed to evaluate the relationship among maternal residual feed intake (RFI) with growth performance and expression of genes involved in lipid metabolism in the offspring.

Line 18: Replace to “Fifty-two cows previously classified as negative or positive RFI by genomic prediction were exposed to fixed-time artificial insemination (FTAI) protocols at 2 and 3 years of age using semen from the same bull.

Line 21: Feedlot or drylot pens?

Line 22: “Cows were weighed every 28 days” for how long?

Line 23 and 24: When?

Line 23 to 25: Sentences could be improved.

Line 25: Why analyzed by experiment? It was never mentioned in the abstract there is more than one study.

Line 26: Year needs to be in the model (if experiment was repeated within years. This information is not clear though).

Line 27: To know if there is an effect of year, it needs to be in the fixed model. According to line 26, year is not in the model. Please, address this discrepancy.

Line 27: Growth, reproductive performance?

Line 31: It seems that the data was analyzed separately. Why?

Line 42 to 43: Poor sentence. It can be more knowledgeable.

Line 46 to 38: Needs to be rewritten. Because of the low nutrient availability?

Line 51: Replace to “can result in impaired growth rates…”

Line 56: and enhance sustainability of production systems.

Line 67: Are authors referring to replacement heifers?

Introduction: This reviewer thinks that in the introduction more information on lipid metabolism needs to be added. For instance, why is your research focused on lipid metabolism?

Line 82: were conducted at the…

Line 95 and 96: How above and below the average. Suggest adding the SD used.

Line 102: Add the days of the experiment (d 0, d 28..)

Were the AI different form exp 1 to exp 2? Details on them and add BW of cows and age in days. How were the cows weighed? Shrunk?

Line 116: The calves were weight at birth and the birth weight was recorded. Suggest changing this sentence.

Line 118 to 120: The weight at 120 days of age (W120) was calculated from the birth weight and weight gain until days of age. Similarly, the weight at 210 days of age (W210) was calculated from the birth weight and weight gain until 210 days of age. The weight or ADG? Authors weighted cattle, so why was weight calculated?

Line 126: Were all the cows in the same pasture?

Line 133: Why were the forage samples divided and only the leaves analyzed? Cows also consume some roots, stem and so on. Hand samples should be collected in a way to represent what is consumed.

149: This methodology does not seem okay because forage mass in one part of the pasture may not be the same in the other area. To estimate forage mass, it is more common to use regression equation using the double sampling technique. Forage availability was calculated as the amount of forage, in kg DM, available 150 within 1 m2 multiplied by 10,000 m2.

150: Why during the second trimester?

Cows were grazing a pasture with less than 6% of CP. No supplementation was provided?

Line 155: How many pens?

Line 168 to 170: This sentence needs to be rewritten. It is not clear.

Line 176: Cow-calf pairs

Line 193: Where is this equation from? Citation needed.

Line 279: Why the P-value is different for gene expression?

Why is the AI in the fixed model? Where the protocols different both experiments? What were the random effects? Why if sex is the fixed term for calf variables why it is not interacting with each other (RFI|SEX)?

297: Deducted instead of deduced?

Why to have a descriptive table?

Table 6: Needs more information. For example, how long was this DMI measured in calves (pre-weaning or post-weaning). Those details need to be in the table.

Reviewer #2: Ms. Ref. No.: PONE-D-21-37544

Title: Performance and muscle lipogenesis in offspring of Nellore cows of different residual feed intake classes

Overview:

This study evaluated cow-calf pairs until weaning in two experiments. Cows with different RFI and different production systems were evaluated. Animal performance and gene expression was used to assessed.

Major comments:

My major consideration in this study is the potential carry-over effect of the higher weight of the cows in the second experiment compared to the first. Although cows were managed with the calves under grazing conditions in the second experiment, body weight could partly compensate nutrition for gestation. This needs to be addressed in the ‘discussion’ section.

Please, provide how ADG was calculated for each animal category.

Please, improve the resolution of figures.

Minor comments:

L056: the citation is incorrect;

L072: the citation is incorrect;

L075-079: in my opinion, the hypothesis and the objectives are somewhat disconnected, especially with the sentence ‘genes related to lipid metabolism’;

L082: ‘conducted at Centro’ instead of ‘conducted Centro’;

L093: provide the country of the company. Please correct this throughout the text;

L107: muscle nomenclature is incorrect;

L108: please provide information of the manufacturer;

L116: ‘The calves were weighed’ instead of ‘The calves were weight’;

L120-121: provide a mean number with standard deviation for calves’ weaning;

L129-130: provide information on how the area was sampled, e.g. which height off the ground, etc.;

L141: the citation is incorrect;

L143: the citation is incorrect;

L143: the citation is incorrect;

L144: the citation is incorrect;

L147: the citation is incorrect;

L158: provide the country of the company;

L201: muscle nomenclature is incorrect;

L208: ‘obtain’ instead of ‘obtained’;

L209: ‘sutured’ instead of ‘closed’;

L210: ‘few days’ is not scientific. Please, be assertive;

L216: muscle nomenclature is incorrect;

L236: ‘totaling’ instead of ‘total’;

L237: the citation is incorrect;

L263: the citation is incorrect;

L264: the citation is incorrect;

L355: muscle nomenclature is incorrect;

L355: ‘from’ instead of ‘to’;

L374: I do not understand what authors mean with this;

L382: muscle nomenclature is incorrect;

L383: ‘from’ instead of ‘to’;

L383: avoid terms such as ‘very important’;

L393: include a reference to support the affirmation;

L394: the citation is incorrect;

L401: the citation is incorrect;

L401-403: rewrite this sentence;

L403: why were you RFI variation lower compared to these other studies? Breed, animals’ weight, and diet were similar?

L403: the citation is incorrect;

L413: the citation is incorrect;

L418-420: what about the physiological differences between cows?;

L426: the citation is incorrect;

L426-427: what were the conditions of the experiment compared to this study?;

L439: the citation is incorrect;

L443: the citation is incorrect;

L444: ‘from’ instead of ‘to’;

L450: the citation is incorrect;

L455: the citation is incorrect;

L466: the citation is incorrect;

L473: avoid terms such as ‘extremely important’;

L512: ‘global’ instead of ‘national’;

L554: avoid using references which are not in English;

L564: check the references. For example, the journal in the citation is not abbreviated.

6. PLOS authors have the option to publish the peer review history of their article (what does this mean?). If published, this will include your full peer review and any attached files.

Reviewer #1: No

Reviewer #2: **Yes: **Rodolpho Martin do Prado

---

## [Author Response · Author response to Decision Letter 0]

13 Apr 2022

Dear Dr. Marcio Duarte,

We appreciated all the efforts you and the reviewers made to improve the quality of this manuscript. In the document attached are the responses for each comment and suggestion made. The changes in the text are highlighted using Word track changes tool in marked-up copy.

Thanks a lot,

The authors

---

## [Decision Letter · Decision Letter 1]

27 May 2022

PONE-D-21-37544R1Performance and muscle lipogenesis of calves born to Nellore cows with different residual feed intake classificationPLOS ONE

Dear Dr. Bonilha,

Thank you for submitting your manuscript to PLOS ONE. After careful consideration, we feel that it has merit but does not fully meet PLOS ONE’s publication criteria as it currently stands. Therefore, we invite you to submit a revised version of the manuscript that addresses the points raised during the review process.

We look forward to receiving your revised manuscript.

Kind regards,

Marcio Duarte, PhD

Academic Editor

PLOS ONE

Reviewers' comments:

Reviewer's Responses to Questions

**Comments to the Author**

1. If the authors have adequately addressed your comments raised in a previous round of review and you feel that this manuscript is now acceptable for publication, you may indicate that here to bypass the “Comments to the Author” section, enter your conflict of interest statement in the “Confidential to Editor” section, and submit your "Accept" recommendation.

Reviewer #2: (No Response)

Reviewer #3: (No Response)

2. Is the manuscript technically sound, and do the data support the conclusions?

Reviewer #2: Yes

Reviewer #3: Partly

3. Has the statistical analysis been performed appropriately and rigorously? 

Reviewer #2: Yes

Reviewer #3: I Don't Know

4. Have the authors made all data underlying the findings in their manuscript fully available?

Reviewer #2: Yes

Reviewer #3: Yes

5. Is the manuscript presented in an intelligible fashion and written in standard English?

Reviewer #2: Yes

Reviewer #3: No

6. Review Comments to the Author

Reviewer #2: Ms. Ref. No.: PONE-D-21-37544R1

Title: Performance and muscle lipogenesis in offspring of Nellore cows of different residual feed intake classes

Major comments:

The manuscript has improved from its early version. However, several issues are pending, some more were raised, and some comments were not addressed at all. For example, the resolution of figures is still the same.

Minor comments:

L019: fifty-two, or fifty-three? Check L094;

L020: it is important to state that two FTAI protocol were used;

L024-25: check this sentence;

L059: there are several entries for ‘body weight’, thus I suggest abbreviating it;

L072: although it was changed, the phrase ‘Results showed that’ could be improved;

L101-106: the manuscript would benefit if authors could clarify as to why three extra cows were added in the second year of the experiment, and also if the 25 cows were all used in the second year;

L129: ‘(i.e. LEA, BF, and RF)’ instead of ‘(LEA, BF, and RF)’;

L133: ‘During pregnancy of year 1’ instead of ‘During pregnancy’;

L133-134: include the pasture management regimen, area and other relevant information, such as fertilization. Also, I suggest clarifying if all cows were kept in the same paddock;

L145: my previous request as completely overlooked, which is frustrating. Provide the country of the company. Please correct this throughout the text;

L145-154: the citations in this paragraph are mostly incorrect. Authors have modified them in the current version, but they are still incorrect;

L146: the numeral should have one space of the degree symbol;

L150: ‘ash’ instead of ‘ashes’;

L155: the term ‘in kg DM’ is incorrect;

L157: ‘t DM/ha’ is incorrect;

L165: provide the full list of information for the system;

L168: how was feed offering adjusted daily if leftovers were not evaluated daily?;

L171: ‘for the leaf fraction’ instead of ‘in the previous item’;

L184: be specific and modify ‘in the previous item’;

L187-191: include the pasture management regimen, area and other relevant information, such as fertilization;

L196: ‘t DM/ha’ is incorrect;

L208: indicate right side of what;

L251: if the work is not published, do not use it as literature, as it was not peer reviewed;

L260: the numeral should have one space of the degree symbol;

L267: if the work is not published, do not use it as literature, as it was not peer reviewed. Furthermore, the text is in a language other than English, which does not fit as a reference for the methodological part of the study;

L345-347: the difference was only for the FABP4 gene, and the others had a tendency. The sentence should be rewritten;

L402-403: the referencing in this sentence is odd;

L415: ‘interferes’ instead of ‘interfere’;

Figure 1: ‘Percent’ instead of ‘Porcent’;

Figure 1: ‘SCD1’ instead of ‘SCD’;

Figure 2: ‘SCD1’ instead of ‘SCD’.

Reviewer #3: This manuscript presents the results from two studies/years conducted to evaluate the effect of maternal RFI on calves’ performance and muscle gene expression. One of the limitations of this study is that the groups of cows were very similar in terms of RFI and the authors observed only a trend for this trait as well as the variables evaluated on claves. I am not convinced that the data was analyzed in the best way. Authors should describe better why FTAI was included in the model. The manuscript language is understandable, but it could be improved.

Overall questions:

1. Was the same group of cows used in both years? Should we consider these as two different experiments or the same experiment repeated for two consecutive years?

2. How can the authors justify the lack of differences between groups for RFI?

Specific comments:

L. 25 – carcass?

L. 25-26 . “Muscle gene expression was evaluated in calves at birth and weaning”

L. 27- Why FTAI is a factor? Was it different?

L. 28-29 – “There was no effect of maternal RFI on calves ADG in the first year.”

L. 36-38 – This conclusion sentence could be more specific to the current study

Introduction – The sentences could be more connected making the text more pleasant to read. I think the introduction could bring some information regarding the heritability of RFI and other characteristics evaluated in this study.

L. 49-50. “received and recorded by the genome” I think this sentence is strange. Please review it.

L.54-56 – This is a long and wordy sentence.

L. 61-65 . Again this is a long sentence and hard to follow. Please review it.

L.96-97. How good is this genomic prediction? Was the RFI evaluated or only predicted?

L. 109 – Ribeye area (REA) and backfat thickness (BFT) is more commonly used for cattle.

L. 113-114. Please, provide more details regarding the equipment and probe used as well as the technician's ability and/or certification to collect and analyze the images.

L. 135. Why not all gestation or the last two trimesters of gestation?

L. 171- Please reword this sentence removing “previous item”

L. 184. Please, be more specific here. What do you mean by “previous item”

Table 1 and 3. Would be possible to concatenate table 1 and 3? DM and TDN are as a % of what?

L. 213- Please delete “An”

L. 217. I suggest deleting “was complete”

L. 220-224. Where were these muscle samples removed from?

L. 228 – What was done with these samples?

L. 275- Again, why FTAI is a factor? Was it different?

L. 275. In my opinion, calves and cow BW should be evaluated as repeated measures.

L. 278. I couldn’t understand it. All data was not normally distributed?

L. 300. Did you evaluate fetal weight?

L. 332- “efficient”

Figures. Please review the axis label “porcent”

Table 7. Where is the DMI data for cows?

L. 400-402. Where is this data? Table 6 is showing a 0.1kg difference for year 1.

L. 400-407. So, how about the current study? Why the difference was so small and only tended to differ?

L. 422 – I think “weaning efficiency” would be a better term than “productive efficiency”. Please consider replacing it across the manuscript.

L. 428-433. This sentence is too long and confusing. Please rephrase it.

L. 444 – These are not body composition data. Please review it throughout the manuscript.

L.471. delete enzyme or protein. It is redundant

L. 488-489. I am not sure if this sentence is accurate. In my understanding, adipogenesis occurs throughout the animal’s life. Please review this sentence.

7. PLOS authors have the option to publish the peer review history of their article (what does this mean?). If published, this will include your full peer review and any attached files.

Reviewer #2: No

Reviewer #3: No

---

## [Author Response · Author response to Decision Letter 1]

11 Jul 2022

We appreciated all the efforts the editor and the reviewers made to improve the quality of this manuscript. In the attached file are the responses for each comment and suggestion made. Thank you.

---

## [Editor Report · Decision Letter 2]

15 Jul 2022

Performance and muscle lipogenesis of calves born to Nellore cows with different residual feed intake classification

PONE-D-21-37544R2

Dear Dr. Bonilha,

We’re pleased to inform you that your manuscript has been judged scientifically suitable for publication and will be formally accepted for publication once it meets all outstanding technical requirements.

Kind regards,

Marcio Duarte, PhD

Academic Editor

PLOS ONE

Additional Editor Comments (optional):

All changes were greatly addressed. The manuscript is now ready for publication.
---

## [Editor Report · Acceptance letter]

21 Jul 2022

PONE-D-21-37544R2 

Performance and muscle lipogenesis of calves born to Nellore cows with different residual feed intake classification 

Dear Dr. Figueiredo Martins Bonilha:

I'm pleased to inform you that your manuscript has been deemed suitable for publication in PLOS ONE. Congratulations! Your manuscript is now with our production department. 

Kind regards, 

on behalf of

Dr. Marcio Duarte 

Academic Editor

PLOS ONE